# Resonant Tunneling Diodes: Mid-Infrared Sensing at Room Temperature

**DOI:** 10.3390/nano12061024

**Published:** 2022-03-21

**Authors:** Florian Rothmayr, Edgar David Guarin Castro, Fabian Hartmann, Georg Knebl, Anne Schade, Sven Höfling, Johannes Koeth, Andreas Pfenning, Lukas Worschech, Victor Lopez-Richard

**Affiliations:** 1Nanoplus Nanosystems and Technologies GmbH, Oberer Kirschberg 4, D-97218 Gerbrunn, Germany; koeth@nanoplus.com; 2Technische Physik, Physikalisches Institut and Röntgen Center for Complex Material Systems (RCCM), Universität Würzburg, Am Hubland, D-97074 Würzburg, Germany; fabian.hartmann@physik.uni-wuerzburg.de (F.H.); georgknebl@gmail.com (G.K.); anne.schade@physik.uni-wuerzburg.de (A.S.); sven.hoefling@physik.uni-wuerzburg.de (S.H.); andreas.pfenning@ubc.ca (A.P.); lukas.worschech@uni-wuerzburg.de (L.W.); 3Departamento de Física, Universidade Federal de São Carlos, São Carlos 13565-905, SP, Brazil; edavidsg89@gmail.com (E.D.G.C.); vlopez@df.ufscar.br (V.L.-R.); 4Stewart Blusson Quantum Matter Institute, University of British Columbia, Vancouver, BC V6T 1Z4, Canada

**Keywords:** resonant tunneling diode, mid-infrared sensing, photosensor

## Abstract

Resonant tunneling diode photodetectors appear to be promising architectures with a simple design for mid-infrared sensing operations at room temperature. We fabricated resonant tunneling devices with GaInAsSb absorbers that allow operation in the 2–4 μm range with significant electrical responsivity of 0.97 A/W at 2004 nm to optical readout. This paper characterizes the photosensor response contrasting different operational regimes and offering a comprehensive theoretical analysis of the main physical ingredients that rule the sensor functionalities and affect its performance. We demonstrate how the drift, accumulation, and escape efficiencies of photogenerated carriers influence the electrostatic modulation of the sensor’s electrical response and how they allow controlling the device’s sensing abilities.

## 1. Introduction

In the last few decades, resonant tunneling diodes (RTDs) have been realized and demonstrated in various materials ranging from semi-conductors to oxides [1,2,3,4,5]. Their spectrum of applications ranges from high-frequency oscillators in the THz range [6] to logic elements [7] and high-sensitive detectors for strain [8], temperature [9], and light [10,11,12,13,14]. Light sensing is especially applied for uses such as molecule and gas spectroscopy, which increasingly demand high-performance devices operating in the mid-infrared (MIR) spectral region. The MIR wavelength range is appealing because several strong absorption lines of important gases, such as CO_2_ (λ=2004 nm, 619 meV), CO (λ=2330 nm), H_2_O (λ=2682 nm), CH_4_ (λ=3270 nm), and HCl (λ=3395 nm, 365 meV), lay within this spectral window [15].

The combination of materials of the so-called 6.1-Å family, such as Ga_0.64_In_0.36_As_0.33_Sb_0.67_ alloys, has been a key ingredient for photodetector applications in the MIR spectral region [16,17]. As a narrow bandgap absorber, this quaternary compound has been routinely used in different applications such as vertical-cavity surface-emitting lasers [18,19], inter-band cascade photodetectors [20,21], or inter-band cascade lasers [16,17]. The used RTD architecture, described in [22], includes an active GaInAsSb layer as an absorber that allows a thorough tuning of the sensor selectivity. In this paper, we present the characterization and explanation of the device operation regimes for optical sensing, in general, and gas sensing applications, in particular, in the MIR spectral region at room temperature. Besides describing the sensor operation at the quantum level and introducing a comprehensive account for its quantum efficiency, the characterization of the sensing functionalities combines complementary experimental approaches that provide the description of both the responsivity and detectivity. We thus determine the noise nature present during the operation of the device. In turn, the theoretical model allows for assessing optimal driving configurations for the sensor sensitivity to changes of the optical inputs, expressed in compact and intuitive mathematical expressions in differential terms with respect to changes of the intensity of the incoming light or varying photon energy. This sensitivity of the RTD also depends on the illumination power, photon energy, and the applied bias voltage [23]. The main goal of this paper is to unveil how the drift, accumulation, and escape efficiencies of photogenerated carriers influence the electrostatic modulation of the sensor’s electrical response and performance.

The transport modulation was assessed experimentally and theoretically for different wavelengths, voltages, and light powers, which allows us to build a grounded understanding of the physical origin of each effect and the contribution of the main structural ingredients to the sensor response. This enables, for instance, our discussion of the role of minority carriers and the optimization of different operating regimes. In addition, the robust resonant transport response of this kind of system has been proven in a wide temperature range [24]. In order to enhance the practical relevance of our findings, the discussion presented here has been focused solely on the room temperature operation of these devices, and all measurements were carried out in normal atmosphere. This has been particularly useful since the effect of water absorption can be used to unveil the sensing abilities of the device within this spectral range in very good agreement with the expected theoretical response.

The core of the device functionalities resides basically in its band structure and doping profile that have been engineered through epitaxial techniques described in the Method Section 2. The resulting conduction band (CB) minimum and valence band (VB) maximum, as well as the doping segmentation, are presented in Figure 1a, along with schematic representations of the band profile under an applied forward bias voltage (middle panel) and the charge accumulation (bottom panel) along the growth direction. These representations take into account the electrons accumulated in the pre-well at the left of the double barrier structure (DBS), $n_{\mathrm{pw}}$ as electron density and the depletion layers with 3D densities, $N_{D1}^{+}$ and $N_{D2}^{+}$.

The charge dynamics that drive the response of the device are schematically represented in the middle panel of Figure 1a, beginning with the photon absorption under illumination. The photogeneration of electron–hole pairs in the absorption layer is followed by the charge drift, the subsequent accumulation of electrons and holes at the right side of the GaInAsSb/GaSb interface and at the DBS, denoted as $n_{\mathrm{ph}}$ and $p_{\mathrm{ph}}$, respectively, and their eventual escape. This leads to an electrostatic tuning of the voltage profile that controls the transport response, and from this point on, our discussion starts with a detailed qualitative and quantitative description of this mechanism.

## 2. Materials and Methods

### 2.1. Design and Fabrication

The diode was grown on an n-type Te-doped GaSb (100) substrate by molecular beam epitaxy with a resonant tunneling structure based on a GaAs_0.15_Sb_0.85_/AlSb double-barrier structure (DBS) [25,26] built in proximity to a lattice-matched Ga_0.64_In_0.36_As_0.33_Sb_0.67_ absorption layer. Such a composition of the quaternary alloy sets a cutoff wavelength of ~3500 nm (350 meV onset energy), suitable for the sensing range for the gases listed before. The details of the fabrication process can be found in [22].

### 2.2. Electro-Optical Readout

The electro-optical transport measurements were carried out with different distributed feedback lasers as a light source supplied by a Pro8000 with LDC8005 laser diode current module. With a Keithley SMU 2400 (Tektronix Inc., Beaverton, OR, USA) as voltage source and current measurement instrument the electrical characteristics were acquired. For measuring the photocurrent, a Lock-In amplifier from Standford Research (SR830) (Stanford Research Systems, Sunnyvale, CA, USA) with a chopper wheel was used. The laser light was collimated with a lens with a focal length of 25 mm. The diodes were contacted with micro-manipulators.

To achieve the responsivity, the power of the laser was measured with a calibrated power meter, and the laser intensity profile was measured with a 6 µm diode and a µm motorized stage in small steps to obtain a high spatial resolution. The volume of the Gaussian profile was then calculated numerically, and the incident light power on the sample corresponds to the volume of the cylinder above the optical active area of the diode. The noise, $I_{n}$, was measured with an ac coupled transimpedance amplifier (Femto DLPCA200) (Femto Messtechnik GmbH, Berlin, DE) and a frequency analyser (Signalhound USB44B) (Signal Hound, Battle Ground, WA, USA) between 100 and 150 kHz with a resolution bandwidth of 10 Hz. As a bias voltage source, a HP Agilent 3245A (Agilent, Santa Clara, CA, USA) universal source was used. The measurement setup and the cables were shielded to reduce the background noise.

For the spectral measurements an IR light source (Newport Corporation, Irvine, CA, USA), a monochromator (Newport Corporation, Irvine, CA, USA) and a voltage source (HP3245A) (Agilent, Santa Clara, CA, USA) to bias the RTD were used. For measuring the photocurrent, a Lock-In amplifier from Standford Research (SR830) (Stanford Research Systems, Sunnyvale, CA, USA) with a chopper wheel was used.

## 3. Results

The effect of the light absorption on the transport properties of the device is exposed in the current–voltage, $I\left(V\right)$, characteristics shown in Figure 1b and obtained after illumination with a laser emitting at an energy of 619 meV. The small current value for RTDs is due to the thick barriers around 4 nm. One could increase the current density by decreasing the barrier thickness, yet too-high current densities lead to a break down due to Joule Heating at higher voltages, which are necessary to fully characterize the diode operation. Furthermore, a complex temperature gradient along the structure might arise, affecting the charge trapping and holes, in particular. Similar structures have attained current densities 2000 A/cm^2^, and the photocurrent characteristics have been discussed by A. Pfenning et al. in [23], yet the RTD in that case had 3 nm thick barriers and a slightly different layout.

The results clearly point out an asymmetric response that depends on the incident optical power density. The current increase for forward bias voltages is related to an electrostatic shift of the voltage drop along the structure produced by the modulation of the charge accumulation [14]. The voltage shift was extracted from the experimental data for an incident optical power density of 79 Wcm^−2^ and is displayed in Figure 1c. Note that there is a point of maximum response with a non-trivial modulation with voltage.

### 3.1. Charge Buildup

To assess the nature of the charge buildup and controlled escape that govern the whole process, a model can be set up by reducing the device response to its main ingredients. Starting with the charge accumulation, it is possible to emulate it, in first approximation, by solving the Poisson equation, $\nabla^{2}V=-\rho \left(z\right)/\varepsilon$, according to the charge profile along the growth z-direction, $\rho \left(z\right)$, as shown in the bottom panel of Figure 1a [27]. If we assume the neutrality conditions, $N_{D1}^{+}l_{1}-\left(n_{\mathrm{pw}}-p_{\mathrm{ph}}\right)=0$ and $N_{D2}^{+}l_{2}-n_{\mathrm{ph}}=0$, where $l_{1}\leq l_{\mathrm{ab}}$ and $l_{2}$ are the lengths of the depletion layers in the absorber and optical window, respectively, and $l_{\mathrm{ab}}$, the length of the absorption layer, then the total voltage drop along the device can be expressed as:(1)$$V_{T}=V_{\mathrm{DBS}}+V_{d}+\Delta V_{\mathrm{ph}}$$

Here, $V_{\mathrm{DBS}}=-n_{\mathrm{pw}}\left(l_{\mathrm{DBS}}+l_{0}\right)/\varepsilon$ and $V_{d}=-n_{\mathrm{pw}}^{2}/2\varepsilon N_{D1}^{+}$, are the voltage drops at the DBS and the absorber in the dark, respectively, with $l_{\mathrm{DBS}}$ and $l_{0}$ as the effective lengths of the DBS and the undoped layer after it. The voltage shift due to photogenerated carriers is given by:(2)$$\Delta V_{\mathrm{ph}}=-\frac{1}{2\varepsilon N_{D1}^{+}}\left(p_{\mathrm{ph}}^{2}-2n_{\mathrm{pw}}p_{\mathrm{ph}}+\frac{N_{D1}^{+}}{N_{D2}^{+}}n_{\mathrm{ph}}^{2}\right)+\frac{p_{\mathrm{ph}}l_{0}}{\varepsilon }$$

By taking into account the fact that the current through the diode is essentially determined by $V_{\mathrm{DBS}}$, the values of $\Delta V_{\mathrm{ph}}$ as a function of the applied forward voltage, V, can be computed from Equation (1) by calculating the amount of $n_{\mathrm{pw}}$ as a function of the total voltage drop $V=-V_{T}$ in dark conditions when $\Delta V_{\mathrm{ph}}=0$:(3)$$n_{\mathrm{pw}}\left(V\right)=N_{D1}^{+}l_{\mathrm{DBS}}\left[\sqrt{1+\frac{2\varepsilon }{N_{D1}^{+}l_{\mathrm{DBS}}^{2}}V}-1\right]$$

The sensor response can be thus characterized by the modulation of $\Delta V_{\mathrm{ph}}\left(V\right)$, which combines the functional dependence on the applied voltage of $n_{\mathrm{pw}}\left(V\right)$, $n_{\mathrm{ph}}\left(V\right)$, and $p_{\mathrm{ph}}\left(V\right)$. In order to describe the relative contribution of just the photogenerated carriers, the mapping of $\Delta V_{\mathrm{ph}}/\left|V_{d}\right|$ as a function of the relative accumulated electrons, $n_{\mathrm{ph}}/n_{\mathrm{pw}}$, and holes, $p_{\mathrm{ph}}/n_{\mathrm{pw}}$, is included in Figure 2a,b for $N_{D1}^{+}=N_{D2}^{+}$ and $N_{D1}^{+}<N_{D2}^{+}$ in panels (a) and (b), respectively. According to the structural parameters of the RTD under analysis, the last term in Equation (2) (proportional to $l_{0}$) provides an insignificant contribution to $\Delta V_{\mathrm{ph}}$, and for this reason, it was neglected in the generation of the color maps in Figure 2a,b. Note that, within this approximation, a positive shift, corresponding to lower absolute values of $V_{T}$, takes place mainly due to a relative increase of trapped holes, and this is reinforced for $N_{D1}^{+}<N_{D2}^{+}$, since the effect of photogenerated electrons is weighted as $n_{\mathrm{ph}}^{2}\;N_{D1}^{+}/N_{D2}^{+}$ in Equation (2). The contours corresponding to $\Delta V_{\mathrm{ph}}=0$ have also been added as reference.

Thus, the efficiency of the photogeneration of these carriers, as well as their subsequent trapping and escape, must complete the picture. The charge dynamics after photon absorption for both electrons and holes are represented schematically in the middle panel of Figure 1a. The hole dynamics can be condensed into three steps. The first one describes the creation of electron–hole pairs after absorption of photons with energy ℏω and controlled by a generation rate, $F\alpha \left(\hbar \omega ,V\right)$, where F=fP/ℏω denotes the photon flux density arriving to the quaternary layer, and $\alpha \left(\hbar \omega ,V\right)$, refers to the ratio between the absorbed and incident light fluxes, which is linearly proportional to the common absorption coefficient. In the former expression, P is the incident optical power density, and f, the fraction of this power transmitted to the sample. The power is measured with a calibrated power meter, while *f* is calculated from the ratio of the volume of the whole laser beam and the volume above the diode. Then, the evolution of the photogenerated hole population, $p_{0}$, can be described as:(4)$$\frac{dp_{0}}{dt}=F\alpha \left(\hbar \omega ,V\right)-\frac{p_{0}}{\tau_{\mathrm{trap}}^{h}}-\frac{p_{0}}{\tau_{\mathrm{miss}}^{h}}-\frac{p_{0}}{\tau_{\mathrm{los}}^{h}}$$

The subsequent steps are characterized by decay terms that take into account the contribution of losses with a rate, $1/\tau_{\mathrm{los}}^{h}$, (not represented in the diagram of Figure 1a), which describes the rate of recombination (radiative or not) of electrons and holes along the absorber region and two paths towards the interface: one driven by a rate, $1/\tau_{\mathrm{miss}}^{h}$, that describes the hole fraction that eventually misses the localization site and, $1/\tau_{\mathrm{trap}}^{h}$, that characterizes the actual trapping. These two are represented in the diagram of Figure 1a. Note that these rates include the drift along the absorption layer; hence, they are affected by the drift length, charge mobility, and local electric field.

After trapping, these holes can subsequently escape through transport channels as described by:(5)$$\frac{dp_{\mathrm{ph}}}{dt}=\frac{p_{0}}{\tau_{\mathrm{trap}}^{h}}-\frac{p_{\mathrm{ph}}}{\tau_{h}}$$

In this case, the time decay is controlled by the lifetime $\tau_{h}$ of the holes.

A symmetric analysis can be applied to the photogenerated, $n_{0}$, and trapped,$\;n_{\mathrm{ph}}$, electrons, yielding analogous equations:(6)$$\left\{\begin{array}{c} \frac{dn_{0}}{dt}=F\alpha \left(\hbar \omega ,V\right)-\frac{n_{0}}{\tau_{\mathrm{trap}}^{e}}-\frac{n_{0}}{\tau_{\mathrm{miss}}^{e}}-\frac{n_{0}}{\tau_{\mathrm{los}}^{e}}\\ \frac{dn_{\mathrm{ph}}}{dt}=\frac{n_{0}}{\tau_{\mathrm{trap}}^{e}}-\frac{n_{\mathrm{ph}}}{\tau_{e}} \end{array}\right.$$

Then, under the stationary condition when, $dp_{0}/dt=dp_{\mathrm{ph}}/dt=dn_{0}/dt=dn_{\mathrm{ph}}/dt=0$, the densities of trapped electrons and holes can be obtained as:(7)$$n_{\mathrm{ph}}\left(V\right)=F\alpha \left(\hbar \omega ,V\right)\eta_{e}\left(V\right)\tau_{e}\left(V\right)\;p_{\mathrm{ph}}\left(V\right)=F\alpha \left(\hbar \omega ,V\right)\eta_{h}\left(V\right)\tau_{h}\left(V\right)$$
where the quantum efficiency terms are given by:(8)$$\eta_{i}=\frac{1}{1+\frac{\tau_{\mathrm{trap}}^{i}}{\tau_{\mathrm{miss}}^{i}}+\frac{\tau_{\mathrm{trap}}^{i}}{\tau_{\mathrm{los}}^{i}}}$$
for electrons and holes, i=e, h, respectively.

Efficiencies of various kinds hold relevant insights about the device performance [28,29] and can be expressed in terms of ratios of concomitant types of decay rates similarly to Equation (8) [30]. In this case, the quantum efficiency weights the ability of the conversion of the incoming light power into effective trapped charges, which are detectable in the current–voltage read-out.

As stated previously, the path towards trapping after photogeneration is a complex process that includes the drift across the absorption layer, yet we are describing it with a single phenomenological term, $1/\tau_{\mathrm{trap}}^{i}$. Thus, this trapping rate can be expressed as an activation probability modulated by the applied voltage:(9)$$\frac{1}{\tau_{\mathrm{trap}}^{i}}=\frac{1}{\tau_{\mathrm{tr},0}^{i}}\exp\left(-\frac{\Delta E-\xi eV}{k_{B}T_{i}}\right),$$
where $\tau_{\mathrm{tr},0}^{i}$ represents a characteristic time that must depend on the layer length and the carrier velocity (mobility and local electric field), while ξ is a leverage factor that relates the local voltage drop with the total applied bias voltage [27], $k_{B}$ is the Boltzmann constant, and $T_{i}$ is the carrier temperature. Moreover, ΔE represents an energy barrier due to built-in electric fields that can, in principle, be screened as the number of photogenerated excitons increases, so that $\Delta E\left(N\right)=\Delta E_{0}/\left[1+\chi \left(N\right)\right]$. Here, $\Delta E_{0}$ is the maximum barrier height, and χ is the electric susceptibility tuned by the density of generated excitons, N, which, within the Clausius–Mossotti approximation [31] can be expressed as $\chi \left(N\right)=pN/\left(1-pN/3\right)$, where p is the exciton polarizability (proportional to the exciton volume) and N is the dipole concentration. In this case, we can assume the density of photocreated excitons as, $N=F\alpha \left(\hbar \omega ,V\right)$.

For the sake of consistency, given that the process characterized by the rate $1/\tau_{\mathrm{miss}}^{i}$ partially coincides with $1/\tau_{\mathrm{trap}}^{i}$, according to the diagram in the middle panel of Figure 1a, we assume a functional dependence on voltage for those paths similar to Equation (9) so that:(10)$$\frac{1}{\tau_{\mathrm{miss}}^{i}}=\frac{1}{\tau_{\mathrm{miss},0}^{i}}\exp\left(-\frac{\Delta E-\xi eV}{k_{B}T_{i}}\right).$$

After these definitions, the quantum efficiency emerges as a logistic function expressed as:(11)$$\eta_{i}\left(V\right)=\frac{\eta_{0}^{i}}{1+\exp\left(-\frac{V-V_{\mathrm{th}}^{i}}{\sigma_{\eta }^{i}}\right)}.$$

This compact expression for the efficiency becomes a figure of merit of the current analysis, with the maximum value determined by:(12)$$\eta_{0}^{i}=\frac{1}{1+\frac{\tau_{\mathrm{tr},0}^{i}}{\tau_{\mathrm{miss},0}^{i}}}<1.$$

In turn, the threshold voltages are defined by:(13)$$V_{\mathrm{th}}^{i}\left(N\right)=\frac{\Delta E\left(N\right)}{\xi e}+\sigma_{\eta }^{i}\ln\left(\eta_{0}^{i}\frac{\tau_{\mathrm{tr},0}^{i}}{\tau_{\mathrm{los}}^{i}}\right),$$
while the steepness of the sigmoid is controlled by the parameter:(14)$$\sigma_{\eta }^{i}=\frac{k_{B}T_{i}}{\xi e}$$
which, under carrier thermalization conditions ($T_{e}=T_{h}$) at room temperature, and considering ξ≈1/2, results in $\sigma_{\eta }^{i}\approx 0.05$ V. Yet, we should note that, under certain excitation conditions, the effective temperature of the transported charge carriers rises above the lattice temperature, with values that might differ between electrons and holes [32]. In this particular region of the diode, however, according to [29], one should expect the lowest possible effective temperature of hot carriers.

Additionally note that the threshold voltage, $V_{\mathrm{th}}^{i}$, in Equation (13) is controlled by two terms: the built-in electric field barriers $\Delta E\left(N\right)=\Delta E_{0}/\left[1+\chi \left(N\right)\right]$, screened in the presence of an exciton gas, and the factor $\sigma_{\eta i}\ln\left(\eta_{0}^{i}\tau_{\mathrm{tr},0}^{i}/\tau_{\mathrm{los}}^{i}\right)$. The latter has a significant contribution in case the parameters in the argument of the logarithm differ in orders of magnitude, turning it positive or negative according to their relative values. For instance, slower drifting (larger $\tau_{\mathrm{tr},0}^{i}$) or efficient losses (smaller $\tau_{\mathrm{los}}^{i}$) lead to an increase of $V_{\mathrm{th}}^{i}$. However, we have assumed that $\eta_{0}^{i}\tau_{\mathrm{tr},0}^{i}/\tau_{\mathrm{los}}^{i}\sim 1$ in the presented simulations, since the screening of built-in electric fields has been confirmed as the leading effect in this case.

The absorption ratio can be simulated as:(15)$$\alpha \left(\hbar \omega ,V\right)=\alpha_{0}\left\{1+\frac{1}{2}\overset{2}{\underset{i=1}{\sum }}\mathrm{erf}\left[\frac{\hbar \omega -\left(E_{g,j}-\gamma V^{2}\right)}{\sqrt{2}\sigma_{\alpha }}\right]\right\}$$
that describes the absorption of photons with energy ℏω above the bandgap of the absorption layer with an energy gap, $E_{g,1}=376$ meV, and in the GaSb optical window with $E_{g,2}=727$ meV. The Stark shift under an applied bias voltage is controlled by the parameter γ [33], $\sigma_{\alpha }$ accounts for the broadening of the absorption function due to homogeneous and inhomogeneous effects, and $\alpha_{0}$ is an intensity ratio which refers to the ratio between the absorbed and incident light fluxes.

In turn, electron and hole lifetimes, $\tau_{e}\left(V\right)$ and $\tau_{h}\left(V\right)$, can be defined in terms of their transmission rates $1/\tau_{i}\left(V\right)\propto {ℑ}_{i}\left(V\right)$. In the case of photogenerated holes, it is unavoidable to analyse their transport through the DBS at forward bias voltages. In this configuration, the contribution of resonant channels for holes must be assessed, and they are emulated by using the transfer matrix approximation [1] as a function of the voltage drop at the DBS, $V_{\mathrm{DBS}}$, as displayed in Figure 2c. A low temperature was used in the simulation in order to better determine the positions of the resonant channels for holes that were obtained at $V_{\mathrm{DBS}}^{\mathrm{res}1}=0.20$ V and $V_{\mathrm{DBS}}^{\mathrm{res}1}=0.30$ V. According to Equation (1) and considering $l_{\mathrm{DBS}}=19$ nm, $l_{0}=2.5$ nm, $\varepsilon =15\;\varepsilon_{0}$, and $N_{D1}^{+}=0.6\times 10^{17}$ cm^−3^, these voltages correspond to total applied voltages of $V_{1}=0.80$ V and $V_{2}=1.65$ V, respectively. The values are close to the nominal values described elsewhere [22]. This allows describing the holes transmission rate across the DBS as a combination of a resonant:(16)$${ℑ}_{h}^{r}\left(V\right)=I_{h}^{r}\overset{2}{\underset{j=1}{\sum }}\exp\left[-\frac{{\left(V-V_{j}\right)}^{2}}{2\sigma_{j}^{2}}\right]$$
and a non-resonant channel:(17)$${ℑ}_{h}^{\mathrm{nr}}\left(V\right)=I_{h}^{\mathrm{nr}}\exp\left(\beta_{h}V\right)$$
with intensities $I_{h}^{r}$ and $I_{h}^{\mathrm{nr}}$, respectively. Here, $\sigma_{j}$ is the transmission peak broadening and $\beta_{h}$ is an escape rate of holes through the DBS. Consequently, the total holes transmission rate is ${ℑ}_{h}\left(V\right)={ℑ}_{h}^{r}\left(V\right)+{ℑ}_{h}^{\mathrm{nr}}\left(V\right)$, all plotted in Figure 2d.

In the case of accumulated electrons at the interface between the absorption layer and the optical window, we may assume just the contribution of non-resonant escape:(18)$${ℑ}_{e}\left(V\right)=I_{e}^{\mathrm{nr}}\exp\left(\beta_{e}V\right)$$
also plotted in Figure 2d, with $\beta_{e}$ as an escape rate of electrons through the interface. In these calculations, we considered the electron and hole lifetimes as $\tau_{i}\left(V\right)\sim 10^{-6}\;s$, taking into account we assumed, and according to carrier lifetime measurements (not presented here) and the lifetimes reported in previous works [9,23], the electron and hole lifetimes were taken as $\tau_{i}\left(V\right)\sim 10^{-6}\;s$. Consequently, we assumed $I_{h}^{r}=1\;{\mu s}^{-1}$ and $I_{h}^{\mathrm{nr}}=I_{e}^{\mathrm{nr}}=0.25\;{\mu s}^{-1}$. The values for the other parameters were $\sigma_{1}=0.29$ V, $\sigma_{2}=0.50$ V, $\beta_{h}=2.5$ V^−1^, and $\beta_{e}=3.7$ V^−1^.

### 3.2. Sensor Read-Out

The calculated trapped densities as a function of voltage are displayed in Figure 2e. The parameters used in these simulations, in addition to the parameters already mentioned, were: $N_{D2}^{+}=1\times 10^{18}$ cm^−3^, $\sigma_{\alpha }=20$ meV, γ=20, and $\Delta E_{0}/\xi e=0.70$ V. Thereby, for high and low optical power densities, the product $f\alpha_{0}\eta_{0}^{i}$ was estimated to be $\sim 10^{-5}$ and $\sim 10^{-8}$, while pN was assumed of the order of $\sim 10^{2}$ and $\sim 10^{-1}\;\Omega^{-1}$, respectively, with p as the exciton polarizability and N is the density of photocreated excitons. The simultaneous electron and hole increase at lower voltages is triggered by the onset of the photoconductivity at the absorption layer, which is a combination of the absorption coefficient and the quantum efficiency. As the voltage grows, this competes with the eventual escape of the accumulated charges, as the de-trapping channels become more active. This balance peaks at point A highlighted in Figure 2e.

In order to correlate the photogenerated charge dynamics with the voltage shift, the path followed by the relative values of the accumulated charges as the applied voltage grows is plotted as a black dotted line over the contour colour map of the voltage shift in Figure 2b. The corresponding voltage shift is displayed in Figure 2f, where, besides the maximum value at V=0.44 V (point A), a dip appears at V≈0.90 V (point B), which is ascribed to the position of the first resonant channel for holes. It is worth noting that the crossing of $n_{\mathrm{ph}}\left(V\right)$ and $p_{\mathrm{ph}}\left(V\right)$ is just a coincidence due to the parameters used in the simulation and plays neither a role in the qualitative picture nor in the position of the dip of $\Delta V_{\mathrm{ph}}$. The experimental values for $\Delta V_{\mathrm{ph}}\left(V\right)$ have also been added into Figure 2f for comparison.

The electrostatic effect induced by charge accumulation also points to two different working regimes of the device for low- and high-power densities of the incident light, which are detailed in Figure 3, where panels (a) and (b) show the values of $\Delta V_{\mathrm{ph}}$ as a function of the applied voltage and the incident photon energy, for high and low optical powers, respectively. For the high-power regime and for ℏω=619 meV, pictured as the upper white dot-dashed line in (a), the calculated $\Delta V_{\mathrm{ph}}\left(V\right)$ in (c) exhibits a maximum at V=0.44 V (peak A) and a second peak at V≈1.00 V (peak B).

We can now correlate the theoretical considerations with the photocurrent defined as the difference between the measured current under illumination and in dark conditions for the same voltage, $I_{\mathrm{ph}}=I_{\mathrm{illumination}}\left(V\right)-I_{\mathrm{dark}}\left(V\right)$. For the experimental data in (f), these peaks are observed at V=0.52 and 1.30 V, respectively. Here, the optical power density is 67 Wcm^−2^, and the photocurrent was normalized to enable comparison. For an excitation energy of ℏω=365 meV, which is below the absorber bandgap of $E_{g,1}=376$ meV, peak B is more pronounced than peak A, as presented in (d). The measured normalized photocurrent in (g) also pictures a reduction of peak A regarding the peak B. The absorption for photon energies below the bandgap can be triggered at higher voltages due to the Stark effect, which reduces the effective energy gap with increasing V, as described in Equation (15). As pointed out before, the non-monotonic behavior of $\Delta V_{\mathrm{ph}}\left(V\right)$ is produced by the balance of trapping and escape efficiencies. The fast increase in the voltage shift for low voltages is due to the onset of the photoconductivity at the absorption layer, which in turn depends on the absorption coefficient and the steepness of the quantum efficiency, discussed above. The slow decrease of the voltage shift for higher voltages is tuned by the escape rates of the holes through the DBS.

For the low-power regime and ℏω=619 meV, the simulation shows, in Figure 3e, the peak B at the same voltage position as for the high-power regime. Experiments carried out with the same incident photon energy, with intensities of $5.6\times 10^{-3}$ Wcm^−2^ and $67\times 10^{-3}$ Wcm^−2^, reveal that peak A is still detected but less pronounced than peak B, as displayed in Figure 3h. The shift of the maximum of $\Delta V_{\mathrm{ph}}\left(V\right)$ towards higher voltages can be explained by a reduced screening of the threshold voltage $V_{\mathrm{th}}\left(N\right)$ as the density of photogenerated excitons decreases. This tunes the quantum efficiency according to Equation (13), producing the shift of the maximum observed in the figures as the screened effective barrier is increased.

The sequential stacking of layers of the RTD architecture (each with a particular functionality) is the main building block for the electrostatic tuning of its transport response under illumination. In that respect, the electron accumulation, described through Equation (3), weights the voltage shift of Equation (2), increasing the contrast of the photoresponse according to the model presented. This effect can be boosted by deeper pre-wells that guarantee more effective electron trapping or by reducing the doping density. Yet, according to the same model, this translates into larger total voltages for the current onset that could become a handicap for certain applications.

Additionally, the trapping efficiency of photogenerated carriers, as described by Equation (9), is sensitive to power changes, explaining the contrasting response for the high- and low-power regimes. This could be considered both an advantage, for intensity modulation sensing, and a disadvantage if a power dependent operation window is an undesirable property.

The sharp voltage tuning at high temperatures is a signature of this photoresponse, and this quality is driven, according to Equation (14), by the ratio between the temperature and the leverage factor ξ that relates the local voltage drop at the absorption layer with the total applied bias voltage (Equation (1)). Thus, since the maximum value for the leverage factor is ξ = 1, the maximum efficiency sharpness is limited by $\sigma_{\eta }^{min}=k_{B}T/e\;$~ 26 mV at room temperature. In turn, if reducing the values of the threshold voltage onset of the photoresponse is an advantage for certain applications, slower drifting (larger$\;\tau_{tr,0}^{i}$) or efficient recombination losses (smaller $\tau_{los}^{i}$) become weaknesses according to Equation (13). Thus, high mobility (determined by high crystal quality, for instance) combined with the size of the absorber layer become relevant targets to aim at.

In order to provide standards for assessing the quality of the sensor response, the responsivity and detectivity have been obtained. The responsivity, $ℜ=I_{\mathrm{ph}}/P$, is determined by the ratio of the measured photocurrent,$\;I_{\mathrm{ph}}$, and the power, P. The laser power was measured as described in the Method Section 2. In turn, the detectivity over a sensor area, A, was obtained as,$\;\;D^{*}=ℜ\times \sqrt{A}/I_{n}$, by measuring the noise spectral density, $I_{n}$, with an ac coupled transimpedance amplifier and a frequency analyzer between 100 and 150 kHz, with a resolution bandwidth of 10 Hz. The measured noise spectral density is displayed in Figure 4a. The fluctuating region close to −0.6 V is produced by the condition of negative differential resistance. The nature of the measured noise can be assessed by emulating the contribution of two sources: Johnson and shot noise. The resulting noise density can be calculated by using the measured RTD current voltage, $I\left(V\right)$, characteristics as $I_{n}\left(V\right)=\sqrt{4k_{B}T\nu /R\left(V\right)+2eI\left(V\right)\nu }$, where ν is the bandwidth in Hz and $R\left(V\right)=dI/dV$ is the differential resistance [34,35]. The blue dashed line in Figure 4a is the calculated noise$\;I_{n}$, while the solid blue line corresponds to the measured values. The good agreement between the experiment and the calculations shows that, indeed, shot and Johnson noise within the −2 V to 2 V range are the main contributions. The corresponding responsivity and detectivity, determined by using the measured noise density, are represented in Figure 4b with peak values of 0.97 A/W and $9.3\times 10^{7}\;\mathrm{cm}\sqrt{\mathrm{Hz}}/W$, respectively, obtained within the 0.5–1 V range.

To be able to better assess the performance of this device, one can compare the detectivity with a state-of-the-art mercury cadmium telluride (MCT) detector from Vigo. Vigo provides a value for a photoconductive used detector of 1.4 × 107 cm√Hz/W at a voltage of 0.4 V and room temperature. A disadvantage of current detectors in the MIR range is the need for low temperatures to achieve high detectivity values. Other detector platforms, which are based on a Type-2 superlattice such as Interband Cascade Detectors [36] or xBn [37], with a contact layer x, a barrier layer B, and an n-type or p-type photon absorbing layer, are difficult to grow because of the hundreds of thin layers that require a long growth time, and each layer or interface must be maintained in good quality, and the strain must be compensated [38,39]. The RTD approach is convincing by its shorter growth time, just one quaternary absorber, and fewer interfaces. Additionally, a RTD has an internal amplification factor which can be up to 1000 at considerably low voltages (below 2 V), lower than those found in an avalanche photodiode [12]. Furthermore, our operating voltage being below 1V is a significant advantage for mobile applications where low power consumption is required.

A complementary tool for the characterization of the sensor response can be extracted from the theoretical model in terms of its relative sensitivity to changes of the input source determined by the light intensity and energy, F and ℏω, respectively. This sensitivity of the voltage response to changing inputs can be determined with respect to the total voltage and defined as:(19)$$S_{\hbar \omega }=\frac{1}{V}\frac{\partial \Delta V_{\mathrm{Ph}}}{\partial \hbar \omega },S_{F}=\frac{1}{V}\frac{\partial \Delta V_{\mathrm{Ph}}}{\partial F}$$
in terms of the partial derivatives of $\Delta V_{\mathrm{Ph}}$, with respect to F or ℏω. Note that, unlike the responsivity, defined as the ratio of output and the strength of the optical input, the functions in Equation (19) characterize the relative strength of changes of the sensor output under fluctuations of the optical inputs.

Defined this way, these functions are ruled by the electronic structure and internal timescales. Thus, the values of $S_{\hbar \omega }$ as a function of the incoming energy and applied voltage are displayed in Figure 4c,e for high- and low-power regimes, respectively. The high-power regime has 1000 times more power than the low-power regime. It is clear that the sensor sensitivity changes as a function of the incoming photon energy and the position of the onset of the absorption, determined by, $E_{g,j}-\gamma V^{2}$ (*j* = 1,2), according to Equation (15), denoting the effective bandgap of GaSb and of the quaternary absorber, respectively. A change in sensitivity occurs close to the bandgaps $E_{g,j}\;j=1,2$. For photon energies between the bandgaps, $\alpha \left(\hbar \omega ,V\right)\;$does not change, and therefore, neither does the sensitivity. The voltage regime with higher sensitivity is determined by the quantum efficiency and the screening of the threshold voltage as the incident power grows, following Equation (12). This leads to the sensitivity shift towards lower voltages in the high-power regime.

In turn, the relative sensitivity with respect to the photon flux density, $S_{F}$, allows for assessing the sensor ability for resolving absorption lines within the spectral band determined by the energy profile of the absorber and the optical window. Panels (d) and (f) in Figure 4 map the performance of $S_{F}$ and show again the contrast between the low and high incoming power regimes. As noted previously, the sensor response is asymmetric with respect to the applied voltage position of the maximum sensitivity. The sharper sensitivity increase at lower voltages is related to the steepness of the quantum efficiency onset that controls the photoconductivity of the absorber layer and, according to Equation (14), is determined by the carriers’ temperature and local leverage factor. In turn, the position of this onset is tuned by the susceptibility increase at higher powers, following Equation (13). In turn, the softer sensitivity decrease at higher voltages is controlled by the escape rate of holes through the DBS.

Charge accumulation controls the voltage tuning and the contrast of the photoresponse. According to the model, the absolute value of the voltage shift increases with increasing light power within a well-defined voltage and photon energy window. The responsivity voltage window is determined by the current onset at a voltage minimum mainly determined by the quantum efficiency for trapping photogenerated carriers and decreases once the trapped carriers escape either by resonant tunneling out or through non-resonant thermionic processes. The absolute value of the responsivity is related to the maximum voltage shift, which is determined by the incident power, as the external drive, and the number of accumulated electrons in the pre-well and doping profiles, defined by internal parameters.

In turn, the sharp detectivity increase, and its maximum value, shown in Figure 4b, which is affected by a differential resistance factor, is weighted by the absorption coefficient (that defines the photon energy onset) and is modulated by the sharpness of the quantum efficiency increase (that controls the voltage shift onset) described in Equation (14).

To evaluate the ability of the RTDs for gas sensing, we used a broadband IR light source dispersed by a monochromator. As stated in the introduction, this measurement is done in normal atmosphere where the absorption line of water is unavoidably present at 2686 nm.

The incident photon flux density profile of the light source in Figure 5a was emulated as:(20)$$I\left(\hbar \omega \right)=\frac{1}{2}\left[1+\mathrm{erf}\left(\frac{\hbar \omega_{c}-\hbar \omega }{\Gamma }\right)\right],$$
with a cutoff energy of $\hbar \omega_{c}=720$ meV and Γ=110 meV. In turn, the absorption ratio for H_2_O was simulated as:(21)$$\alpha_{H_{2}O}\left(\hbar \omega \right)=\frac{1}{5}\exp\left[-\frac{{\left(\hbar \omega -E_{H_{2}O}\right)}^{2}}{2\sigma_{H_{2}O}^{2}}\right]$$

Here, $E_{H_{2}O}=450$ meV, and $\sigma_{H_{2}O}=25$ meV [15]. Thus, the effective incoming photon flux density in this case can be defined as:(22)$$F\left(\hbar \omega \right)=I\left(\hbar \omega \right)-\alpha_{H_{2}O}\left(\hbar \omega \right).$$

Using these considerations, the resulting calculated map for $\Delta V_{\mathrm{ph}}\left(V,\hbar \omega \right)$ is displayed in Figure 5b. Figure 5c,d picture the measured photocurrent as a color-gradient map and as a cross-sectional view, respectively. The simulated voltage shift fits well to the measured values. The small dip at 329 meV is due to a reduction in the light power of the source and was consequently not included in the simulated data. The change in light power was cross-checked with a commercially available photodiode (Hamamatsu P11120-201). In contrast to the dip, the peak at 710 meV is due to the already mentioned absorption in the GaSb layer. The simulated and measured absorption line of water is well pronounced and confirms that this RTD can be used as a detector for gases in the MIR regime. In particular, the two operating points for different powers and energies allow the optimum operating regime to be selected at all times. In Figure 3f,g, one can see the evolution of the two peaks at 0.52 V and 1.3 V as the power regime changes. Then, according to Figure 4e,f, the low voltage peak is more suitable to detect intensity changes in the high-power regime, while the second would be more useful at low powers.

## 4. Conclusions

In summary, we have been able to characterize the main ingredients that control the operation of RTD photosensors, as well as their photoresponse. The role of minority carriers has been correlated to the trapping efficiency and quantum transmission of photogenerated electrons and holes and the tuning of these effects with external bias and the parameters of the incoming light. The efficiency of the minority carriers trapping enhances the sensing abilities. The sensor quality has been assessed in terms of the responsivity and detectivity. The model allowed us to define the relative sensitivity as an additional ingredient for the characterization of the sensor operation pointing to two different regimes that can be traced by varying the intensity of the photoexcitation. Finally, the sensor response to the presence of H_2_O molecules at room temperature has been unambiguously demonstrated in normal atmosphere.

## Figures and Tables

**Figure 1 nanomaterials-12-01024-f001:**
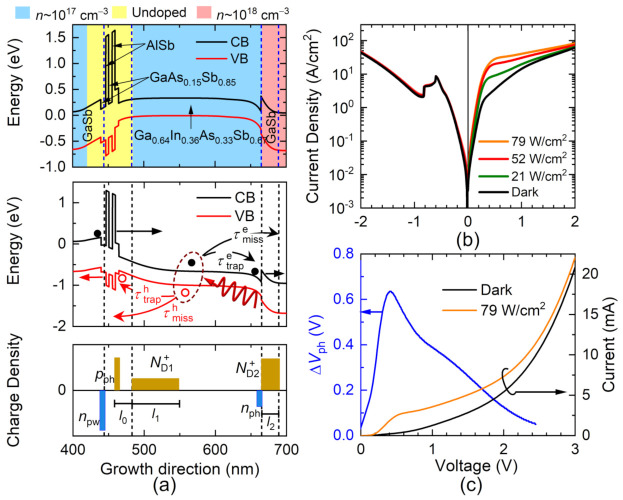
(**a**) Upper panel: Band structure and doping profile of the RTD. Middle panel: Representation of the band structure under an applied forward voltage and illustration of the charge carrier photogeneration, drift, trapping, and escape. Lower panel: Scheme of the charge density distribution. (**b**) Current–voltage characteristics of forward and reverse bias voltage in the dark (black line) and under illumination of various optical powers (coloured lines) using an incident light of 619 meV. (**c**) Current–voltage characteristics under dark and illumination conditions and the extracted value of the voltage shift (blue line) under illumination for an incident light source of 79 Wcm^−2^ and 619 meV.

**Figure 2 nanomaterials-12-01024-f002:**
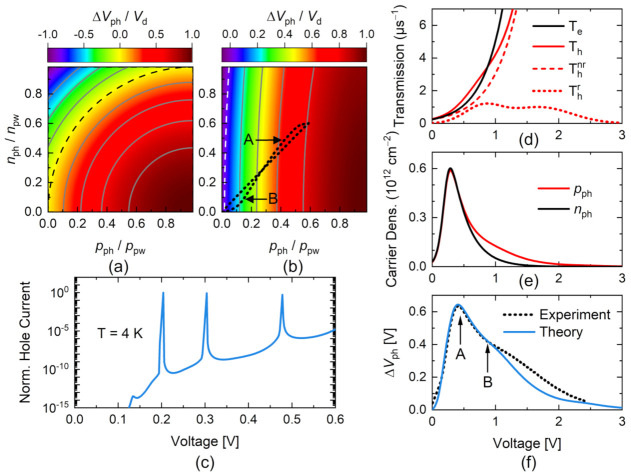
Calculated relative voltage shift, $\Delta V_{\mathrm{ph}}/\left|V_{d}\right|$, as a function of the photogenerated carriers for: (**a**) $N_{D1}^{+}=N_{D2}^{+}$ and (**b**) $N_{D1}^{+}<N_{D2}^{+}$. Dashed black contour lines represent $\Delta V_{\mathrm{ph}}=0$. (**c**) Calculated hole current as a function of the bias voltage at the DBS, $V_{\mathrm{DBS}}$. (**d**) Emulated electron (black line) and hole (red line) transmission rates. The latter was obtained from resonant (dotted line) and non-resonant (dashed line) contributions. (**e**) Calculated trapped electron (black line) and hole (red line) densities as a function of the applied voltage. Changes on these carrier densities are also represented in (**b**) by a black dotted line. (**f**) Calculated (blue solid line) and measured (black dotted line) voltage shift, $\Delta V_{\mathrm{ph}}$, for an incident light energy of 619 meV and 79 Wcm^−2^. A and B indicate the positions of the $\Delta V_{\mathrm{ph}}$ maximum and the first transmission peak, at 0.44 and 0.92 V, respectively.

**Figure 3 nanomaterials-12-01024-f003:**
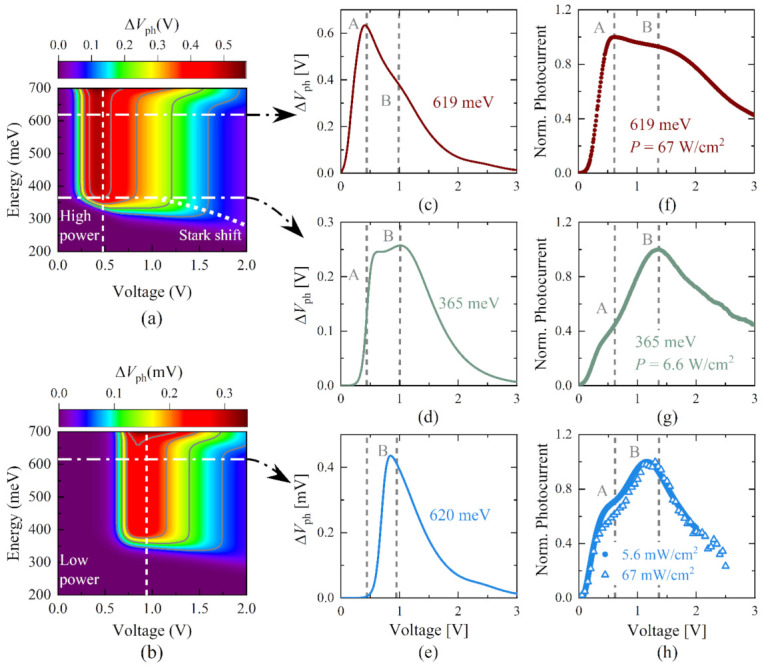
Calculated $\Delta V_{\mathrm{ph}}$ as a function of the applied voltage and incident photon energy for: (**a**) high-power Wcm^−2^ and (**b**) low-power mWcm^−2^ regimes. Horizontal dot–dashed lines indicate the photon energies at which simulations and measurements were performed. Calculated voltage shift for: (**c**) ℏω=619 meV and (**d**) ℏω=365 meV in the high-power regime and for (**e**) ℏω=619 meV in the low-power regime. Normalized photocurrent measured for: (**f**) ℏω=619 meV and (**g**) ℏω=365 meV in the high-power regime and for (**h**) ℏω=619 meV for two power densities, 5.6 mWcm^−2^ (blue dots), and 67 mWcm^−2^ (blue triangles), in the low-power regime. Vertical dashed lines are used as reference to mark the approximate position of peaks A and B.

**Figure 4 nanomaterials-12-01024-f004:**
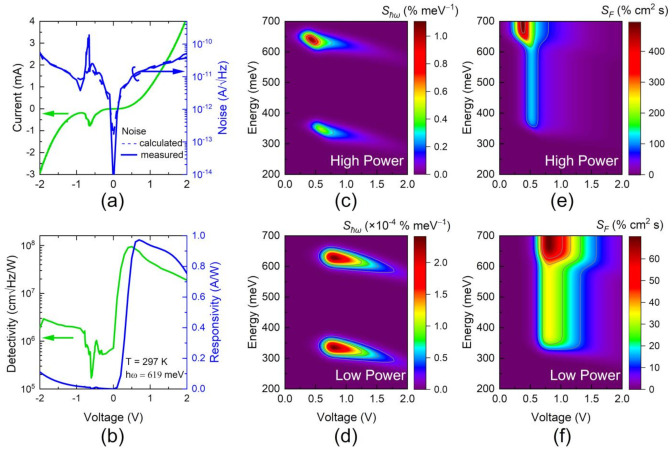
(**a**) Measured (blue solid) and calculated (blue dashed) noise for the current–voltage characteristic (green) curve. (**b**) Responsivity (blue) and detectivity (green) based on the measured noise in (**a**). Calculated relative sensitivity for the energy of the incoming photons, $S_{\hbar \omega }$, in the (**c**) high- and (**d**) low-power regimes. Calculated relative sensitivity for the flux density of the incoming photons, $S_{F}$, in the (**e**) high- and (**f**) low-power regimes.

**Figure 5 nanomaterials-12-01024-f005:**
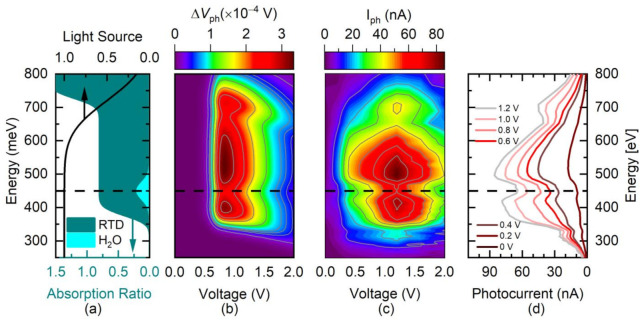
Photoresponse in the low-power regime for a varying excitation profile of the light source. (**a**) Photoexcitation profile of the light source (black line) and light absorption in the diode (dark green) and for the H_2_O vapor (light green). (**b**) Calculated $\Delta V_{\mathrm{ph}}$ using the incoming parameters of panel (**a**). (**c**) Experimental measurements of the photocurrent response within the same range of parameters used in panel (**b**). (**d**) Profiles of the photoresponse for various voltages correlated to the power density of the incoming photons from the photodiode used as light source.

## Data Availability

Data available on request due to restrictions, e.g., privacy or ethical.

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
