# Peer review of "Resonant Tunneling Diodes: Mid-Infrared Sensing at Room Temperature"

_nanomaterials, 2022, doi:10.3390/nano12061024_

Round 1

Reviewer 1 Report

This paper discusses the operation principle and modeling of the RTD-based MIR photodetector. The subject is timely and should interest many researchers in this field. This paper is worth publishing in the Nanomaterials after some revisions. My comments are shown below.

  1. This paper lacks some important discussions. First, it is important to include comparison to other types of MIR detectors, advantages and disadvantages expected from the model discussed in this paper. Also, discussion on the expected maximum performances based on this model should be added.
  2. The current density of the RTD used is approximately 10A/cm^2 as seen in Fig. 1. This is relatively small value as an RTD. Effects of the current density of the RTD structure on the operation of the photodiode should be discussed.
  3. line 179. Here, the authors mention about the tau_miss. I do not understand the difference from the tau_los. Please, explain more details.
  4. This photodetector shows different behaviors for low and high power regimes. This indicates strong non-linearity for a sensor operation. Comments on this for practical applications should benefit readers.
  5. minor points: Figure caption in Fig. 3. “(orange triangles)” should be blue triangles.

Author Response

Please see attachment. The response letter includes the answers to all four reviewers comments.

Reviewer 2 Report

The manuscript reports the fabrication, characterisation, and theoretical modelling of resonant tunnelling diode (RTD) photodetectors with GaInAsSb absorbers at mid-infrared wavelengths (2-4 µm). The experimental characterisation and theoretical analysis presented in the paper are very thorough and give valuable insight into the operation and the physics of these devices. I recommend to publish the manuscript as is.

The only optional revision that I would suggest is to add a discussion of how the performance of GaInAsSb RTDs compare with other types of mid-IR photodetectors, in terms of responsivity, detectivity, operation temperature, spectral bandwidth, etc.

I noticed an oversight in the authors contributions: "formal analysis, X.X."

Author Response

(The authors gave the same response as above.)

Reviewer 3 Report

The article is well written, and scientific soundness is high. The presentation of results and the figure qualities are almost perfect. Taking this into account, I recommend a minor but obligatory revision of the draft.
Comment
In the introduction, part authors describe the state-of-the-art of RTDs for the MIR sensing at room temperatures well, but it will be good that the authors highlight the most significant novelty of the presented research in one or two sentences.

Author Response

(The authors gave the same response as above.)

Reviewer 4 Report

In this manuscript, the authors characterize the main ingredients that control the operation of RTD photosensors, as well as their photoresponse. The role of a few carriers is related to the trapping efficiency and quantum transmission of individual photogenerated electrons and holes, and these effects are related to external bias and incident light parameters. The conjecture is verified in detail in the manuscript by theory and simulation. And at the end, the response of the sensor to the presence of H2O molecules at room temperature has been unambiguously confirmed in a normal atmosphere. Also, the manuscript is well written and the relevant logical and theoretical support is very adequate. Therefore, I recommend this manuscript be published in Nanomaterials. More specific feedback suggestions are given as follows:

  1. The units in the figure do not match the figure caption. There is a format problem in "Wcm-2" described in Figure 1, and there are similar errors in the article. Please check carefully.
  2. The author mentions the evolution of the photogenerated hole population in line 175, followed by a brief explanation of decay terms. However, the explanation of the influencing factors for decay terms mentioned here is not clear enough, and further additions are suggested.
  3. At line number 318 the author suggests the Stark effect, the citation of the theory, is incorrect and suggests providing explanations for the theory.
  4. In the end the author mentions using water to evaluate the ability of the RTDs for gas sensing, could the author provide reasons for choosing water over other gases?
  5. At line number 419 the author mentions " The change in light power was cross-checked with a commercially available photodiode ". Please give relevant references to support this.
  6. In references, the superscript and subscript formats need to be checked and modified.

Author Response

(The authors gave the same response as above.)

Round 2

Reviewer 1 Report

The authors addressed most of the concerns in the previous review. I recommend publication in nanomaterials.